# Anticytokine Autoantibodies in Infectious Diseases: A Practical Overview

**DOI:** 10.3390/ijms25010515

**Published:** 2023-12-30

**Authors:** Rob J. W. Arts, Nico A. F. Janssen, Frank L. van de Veerdonk

**Affiliations:** 1Department of Internal Medicine, Radboud Institute of Molecular Life Sciences (RIMLS), Radboudumc Center for Infectious Diseases (RCI), Radboud University Medical Center, 6525 GA Nijmegen, The Netherlands; nico.janssen@mft.nhs.uk (N.A.F.J.); frank.vandeveerdonk@radboudumc.nl (F.L.v.d.V.); 2Center of Expertise in Mycology Radboudumc, Canisius-Wilhelmina Hospital, Radboud University Medical Center, 6525 GA Nijmegen, The Netherlands; 3Department of Infectious Diseases, The National Aspergillosis Centre, Wythenshawe Hospital, Manchester University NHS Foundation Trust, Southmoor Road, Wythenshawe, Manchester M23 9LT, UK; 4Division of Evolution, Infection and Genomics, Faculty of Biology, Medicine and Health, University of Manchester, Manchester M13 9PL, UK

**Keywords:** cytokine, secondary immunodeficiency, anti-GM-CSF, anti-interferon, anti-interleukin, non-tuberculous mycobacteria, cryptococcus

## Abstract

Anticytokine autoantibodies (ACAAs) are a fascinating group of antibodies that have gained more and more attention in the field of autoimmunity and secondary immunodeficiencies over the years. Some of these antibodies are characterized by their ability to target and neutralize specific cytokines. ACAAs can play a role in the susceptibility to several infectious diseases, and their infectious manifestations depending on which specific immunological pathway is affected. In this review, we will give an outline per infection in which ACAAs might play a role and whether additional immunomodulatory treatment next to antimicrobial treatment can be considered. Finally, we describe the areas for future research on ACAAs.

## 1. Introduction

The existence of anticytokine autoantibodies (ACAAs) has been known since the late 20th century when they were first discovered in patients with autoimmune disorders and thymoma [1]. These antibodies are characterized by their ability to target and neutralize specific cytokines (Figure 1). In the meantime, several auto-immune diseases and infectious diseases have been associated with the presence of ACAAs, sometimes in a causative, but most frequently in an associative role [2].

Interestingly, not all ACAAs have neutralizing capacities and titers differ greatly between subjects [3]. Even in a healthy population, individuals with ACAAs can be found [4]. In a Danish study, in almost 9000 blood donors, 86% of the participants demonstrated at least one ACAA, which shows that low titers of ACAAs are relatively common, but also major differences between different types of ACAAs were observed. ACAAs against IL-6 were most common (65%), compared to only 10% of the participants showing antibodies against GM-CSF [4]. It is believed that in healthy individuals ACAAs play a role in fine-tuning the immune response, ensuring proper regulation of cytokine activity during disease. After infection, antibody titers slowly decline and might even become negative again. For anti-GM-CSF autoantibodies, it has been shown for example that only high titers correlate with the development of pulmonary alveolar proteinosis and susceptibility to opportunistic infections [5,6]. However, in a follow-up study on Danish blood donors, it was shown that even in this healthy population the cumulative presence of ACAAs correlated with several proxies for immune function, such as a self-reported health score and antibiotic prescriptions [7].

The presence of ACAAs has been implicated in a wide range of diseases, including autoimmune disorders such as rheumatoid arthritis, systemic lupus erythematosus, and psoriasis [2]. Additionally, ACAAs have been associated with immunodeficiencies and infectious diseases, where their presence can impact the immune response against pathogens. Understanding the mechanisms behind the production and function of ACAAs is an active area of research, as it holds promise for the development of novel therapeutic interventions [2]. By targeting these autoantibodies or inhibiting their production, the delicate balance of cytokines could potentially be restored, which could serve as additional treatment in certain infectious diseases.

In this review, we will summarize per infectious disease which ACAAs have been associated with them, in which patients’ ACAA diagnostics should be considered, and which potential treatment options have been explored thus far. Here, we will provide a practical guide for the clinician and point out potential research interests for future research.

## 2. Bacterial Infections

### 2.1. Tuberculosis

Interferon-gamma (IFNγ) has been known to play a central role in the immune defense against *Mycobacterium tuberculosis*. Given this central role, in 1998, the presence of IFNγ auto-antibodies was already examined. In this study, it was shown that neutralizing anti-IFNγ auto-antibodies that were present in 10 out of 30 HIV-negative tuberculosis (TB) patients, with the highest antibody titers found in patients with advanced disease, and those with detectable IFNγ in serum [8]. In a cohort of 74 HIV-negative patients with anti-IFNγ auto-antibodies from the USA and Thailand, with almost all the American patients being of Asian descent, 5 diagnoses of tuberculosis were made (two pulmonary and three disseminated cases) [9]. In addition, several case reports demonstrate the presence of anti-IFNγ auto-antibodies in disseminated tuberculosis, often in Asian patients [10,11,12]. In contrast to these findings, in a large cohort study performed in Thailand and Taiwan, only 1/9 patients with disseminated TB and 1/49 patients with pulmonary TB demonstrated the presence of anti-IFNγ auto-antibodies, which was not significantly different from the frequency in healthy controls (1/48). However, these auto-antibodies were present significantly more frequently in patients with disseminated non-tuberculous mycobacterial (NTM) infections in this study [13]. Furthermore, in a Japanese study, no anti-IFNγ auto-antibodies were found in 189 pulmonary TB patients [14]. Therefore, it remains to be determined how extensive the role of anti-IFNγ auto-antibodies is in TB, especially in disseminated disease.

Autoantibodies to other potentially relevant cytokines in the immune response to TB, such as interleukin (IL)-12 or granulocyte-macrophage colony-stimulating factor (GM-CSF), have so far only rarely been assessed in individual patients in case reports [11,15,16,17], but never systematically in larger cohorts of patients. The same holds true for potential therapeutic interventions targeting ACAAs in TB. Thus far, only case reports have described attempted treatment with IFNγ supplementation, rituximab, or cyclophosphamide combined with prednisone in patients with anti-IFNγ auto-antibodies and a double infection with TB and NTM with both positive and negative outcomes [9,18,19,20].

The main research aim for ACAAs in TB would therefore be to first define whether they are functionally neutralizing IFNγ and to assess their role in pulmonary and disseminated disease, for which a cohort analysis would be beneficial.

### 2.2. Non-Tuberculous Mycobacterial Infections

The role of anticytokine autoantibodies (ACAAs) in NTM infections, especially anti-IFNγ auto-antibodies, has been widely studied. Most patients are adults, but also in children, rare cases have been reported [21,22]. Apart from multiple case reports, several larger cohort studies have been performed. Two systematic reviews provide a good overview of patients with anti-IFNγ auto-antibodies. Of all patients with anti-IFNγ auto-antibodies 83.5% and 49.2%, respectively, presented with an NTM infection [20,23]. In 52 patients with disseminated NTM infection in Thailand and Taiwan, 81% demonstrated anti-IFNγ auto-antibodies [13]. In a Taiwanese study with patients with disseminated NTM infections, 45 of 46 patients had anti-IFNγ auto-antibodies [24] and in another Thai study, in all 19 patients with disseminated NTM infection, these auto-antibodies were found [25]. In an American study with 35 patients with disseminated or pulmonary NTM infections, 17% (6/35) of patients demonstrated a high titer of anti-IFNγ auto-antibodies with proven neutralizing capacity [26]. In the previously mentioned cohort study of 74 HIV-negative patients with anti-IFNγ auto-antibodies from the USA and Thailand, 67 were diagnosed with a disseminated NTM infection, the majority of which involved *Mycobacterium avium* complex (MAC) in the USA and *Mycobacterium abscessus* in Thailand [9]. In a Japanese study with 91 patients with pulmonary and 51 with disseminated NTM infections, anti-IFNγ auto-antibodies were found in 37 patients, with neutralizing capacity shown in 31, all of whom had disseminated NTM infection [14]. In 135 Australian patients with a pulmonary NTM infection due to *Mycobacterium avium*, *intracellulare*, *kansasii*, or *lentiflavum*, significantly increased titers of anti-IFNγ and anti-GM-CSF auto-antibodies were found compared to controls and patients with a *Mycobacterium abscessus* infection. Anti-IFNγ and anti-GM-CSF auto-antibody titers did not correlate, therefore constituting separate risk factors [27]. Anti-GM-CSF auto-antibodies were also found to predispose to pulmonary NTM infection in pulmonary alveolar proteinosis (PAP, which is caused by these anti-GM-CSF auto-antibodies). However, whether this is due to the ACAAs, or to the altered pulmonary architecture (or both) remains elusive [27,28,29,30]. In conclusion, although the prevalence of ACAAs varies greatly between different cohorts and not all studies specify whether ACAAs found to have functionally neutralizing capacity, anti-GM-CSF and especially anti-IFNγ auto-antibodies have been shown as a risk factor for the development of (especially disseminated) NTM infections. Anti-IFNγ auto-antibodies are not routinely investigated in all patients with NTM infections due to technical challenges. A convenient workaround could be to perform an interferon-gamma release assay (IGRA): this test would be indeterminate in the presence of anti-IFNγ auto-antibodies, as the positive control (a mitogenic stimulus) would turn out negative due to the antibodies [31].

Therapeutic options targeting ACAAs in NTM infection have only been explored and presented in case reports or case series so far [20]. Here, we will summarize the most important findings. First, IFNγ supplementation has shown both positive and negative outcomes, with no clear predictors for therapy failure [18,20,32,33]. Second, there are two reports of intravenous immunoglobulin (IVIG) therapy in patients with anti-IFNγ auto-antibodies. One describes a 44-year-old female with disseminated MAC infection who received IVIG with good clinical effect after 5 months of antibiotic treatment and insufficient clinical improvement [34] and the other describes a 24-year-old girl with disseminated *M. simiae* infection and low immunoglobulins, in whom preemptive IVIG therapy was directly started [35]. Third, a combination of plasmapheresis and cyclophosphamide was effective in a 38-year-old female Filipino patient with MAC osteomyelitis during a 3-year-follow-up. The antibody titer decreased and clinical symptoms improved, but antimycobacterial treatment had to be continued to prevent relapse [36]. In a cohort study in Thailand, eight patients with anti-IFNγ auto-antibodies and NTM infection were treated with cyclophosphamide and corticosteroids. Five patients achieved remission—of whom two could discontinue antimycobacterial treatment—two relapsed and one died [19]. Fourth and last, in case reports or small case series, the effect of B cell depleting therapy with rituximab has been described in patients with anti-IFNγ auto-antibodies whose infections were refractory to antibiotic therapy, all with good clinical effect and little side effects and resulting in sustained remission [37,38,39,40,41,42,43]. Comparably, four Asian women with treatment-refractory disseminated NTM infections that were also not responsive to additional immunomodulatory therapy with plasmapheresis (1 patient) or IFNγ supplementation (3 patients) responded to treatment with rituximab. All patients demonstrated improvement in the first 2–6 months, but to ensure sustained clinical improvement rituximab was continued for a least a year. No infectious complications were observed [44]. In another cohort of Thai and American NTM patients with anti-IFNγ auto-antibodies, eight received cyclophosphamide and ten received rituximab, both resulting in reduced anti-IFNγ auto-antibody titers, with no remarkable side effects in the patients receiving rituximab [9]. In a Thai study with patients with anti-IFNγ auto-antibodies and difficult-to-treat NTM infections, eleven patients were treated with cyclophosphamide and four with rituximab. Nine out of the eleven patients in the cyclophosphamide group achieved remission compared to all four in the rituximab group, although remission occurred faster in the cyclophosphamide-treated patients (after 9 versus 84 days) [45]. One case report is available on a 31-year-old Filipino woman with disseminated MAC infection which was unresponsive to several lines of antibiotic treatment. Subsequent treatment with rituximab did not reduce disease activity or anti-IFNγ auto-antibody titers, but treatment with daratumumab (anti-CD38) was ultimately effective [46]. The addition of bortezomib (a proteasome inhibitor used in multiple myeloma) to rituximab when rituximab monotherapy is unsuccessful has also been suggested [47]. In another 38-year-old Filipino female patient with anti-IFNγ autoantibodies and disseminated MAC infection, extensive antibiotic treatment, surgical debridement, and rituximab were not effective, but following the addition of bortezomib there was a slow clinical improvement over months [47].

All in all, several attempts of host-directed immune-modulating therapy in patients with NTM infections and anti-IFNγ auto-antibodies have been described, with positive outcomes, especially for rituximab. The time now seems right to perform the first randomized clinical trial to define the exact beneficial effects of immunomodulation in NTM infections.

### 2.3. Nocardia

*Nocardia* infections were first described in association with pulmonary alveolar proteinosis (PAP), which is caused by anti-GM-CSF autoantibodies, in 1960 [6,29,48]. Since then, in a cohort of 104 PAP patients with a mean follow-up duration of 3.4 years, the most commonly identified opportunistic infection was nocardiosis (10%), which was found in the patients with the highest anti-GM-CSF auto-antibody titers [49]. Disseminated *Nocardia* infections are also more common in PAP patients [6,50,51]. However, anti-GM-CSF auto-antibodies have now also been associated with pulmonary and disseminated *Nocardia* infections in patients without PAP. In five patients with disseminated or cerebral abscesses due to *Nocardia*, all patients possessed high titer anti-GM-CSF auto-antibodies [52,53,54]. Apart from anti-GM-CSF auto-antibodies, anti-IFNγ auto-antibodies have also been reported in one case report of a severe disseminated *Nocardia* infection [55].

Therapeutic immunomodulatory options, such as GM-CSF supplementation or rituximab, have not yet been explored in patients with ACAAs in *Nocardia* infection. IFNγ supplementation has been applied in three patients with severe disease, in whom ACAAs were not assessed [56]. The necessity for prophylactic treatment for new *Nocardia* infections or cryptococcosis, screening for recurrence of infection or screening for PAP in these patients remains elusive. Establishing cohorts of non-PAP patients with *Nocardia* infection to assess the presence of anti-GM-CSF auto-antibodies and long-term follow-up data in this group of patients are essential to answer these questions.

### 2.4. Miscellaneous

In literature, three cases of severe staphylococcal infection have been described in the presence of neutralizing anti-IL-6 auto-antibodies, with the observation that low/normal CRP values were measured despite clinically severe disease. The cases concern an 11-month-old boy from Haiti with recurrent cellulitis and abscesses due to *Staphylococcus aureus* [57], a 56-year-old Japanese woman with severe recurrent *Staphylococcus aureus* soft tissue infections [58], and a 20-month-old Czech girl with a presumed *Staphylococcus aureus* septic shock [59]. There is also a fourth case of a 67-year-old Japanese man anti-IL-6 auto-antibodies with no *S. aureus*-related infection. He presented with an empyema due to *E. coli* and *S. intermedius* [17]. Hence, a low CRP in severe disease should result in clinical suspicion of the presence of anti-IL-6 auto-antibodies.

In ten patients with hidradenitis suppurativa (where superinfections with *S. aureus* are common), no ACAAs against IL-1α, IL-6, IL-10, IL-17, or IFNα were found [60].

The literature also provides multiple case reports and patient cohorts in which *Salmonella* infections occur in patients with anti-IFNγ auto-antibodies, consistent with *Salmonella*’s intracellular life cycle. These include a 65-year-old German woman with a disseminated MAC infection who suffered from *Salmonella typhi* sepsis [61], a 45-year-old Thai patient with recurrent mycobacterial infections also suffering from recurrent *Salmonella* infections (which were ultimately controlled with co-trimoxazole prophylaxis) [12]. A Chinese study describes three patients with anti-IFNγ auto-antibodies with recurrent *Salmonella* bacteremia and two other patients with a single infection [62]. A Taiwanese study describes four patients with neutralizing anti-IFNγ auto-antibodies, all of whom developed NTM disease and three of whom developed *Salmonella*-related infections (besides two with systemic *Talaromyces marneffei* infection, one patient with pulmonary TB, one with *Legionella* pneumonia, one with herpes zoster, one with oral herpes simplex and one with Epstein–Barr virus (EBV) associated disease) [63]. Furthermore, in a cohort of 46 patients with NTM infections and anti-IFNγ antibodies, 40% also had a history of salmonellosis [24]. In addition, in several patient cohorts with NTM infections and anti-IFNγ auto-antibodies, coinfection with *Salmonella* has been reported [9,14,19,61,64,65,66]. In some, immunomodulatory therapy has been applied for treatment of the NTM coinfection, but also demonstrated a beneficial effect on the *Salmonella* infection [13,14,19].

Three reports of recurrent or severe *Burkolderia* infections in patients with ACAAs have been reported in the literature. The first one concerns a 45-year-old Cambodian woman with anti-IL-12p70 autoantibodies which resulted in a recurrent *Burkholderia gladioli* lymphadenitis [67]. The second patient had proven neutralizing anti-IFNγ auto-antibodies and developed disseminated infection with *Burkholderia gladioli*, later followed by disseminated infection with *Mycobacterium chelonae*. She died due to sepsis [68]. The third patient also demonstrated anti-IFNγ auto-antibodies and suffered from recurrent infections with different species of NTM, *Talaromyces,* and *Burkholderia pseudomallei* [62].

Additionally, infection with *Legionella* has sporadically been described in patients with the presence of anti-IFNγ auto-antibodies [63,66].

In a cohort of 35 acute respiratory distress syndrome (ARDS) and 13 sepsis patients, 3 patients showed anti-IL-6 auto-antibodies, two anti-IFNω auto-antibodies, two anti-IFNγ auto-antibodies, and one anti-IL-1α auto-antibodies. Unfortunately, these auto-antibodies were not stratified by pathogen, and it was not known whether they were already present before the onset of sepsis [69].

No human data are available on the role of ACAAs in bacterial meningitis. However, in two rat studies with *Haemophilus influenzae* meningitis induced by intraperitoneal injection, it was shown that both in serum and in cerebrospinal fluid, ACAAs against IFNγ and tumor necrosis factor (TNF)-α were induced during the infection [70,71].

## 3. Viral Infections

### 3.1. Herpes Viruses

The most commonly described association of ACAAs with viral infections is with herpes viruses, especially varicella zoster virus (VZV). A 26-month-old girl with partial recombination activating gene (RAG) deficiency suffered from a prolonged VZV infection with proven auto-antibodies against IL-12, IFNα, and IFNω [72]. Anti-IFNα and IFNω auto-antibodies were also described in an 18-year-old female with common variable immune deficiency and severe facial herpes vegetans with recurrent herpes simplex viremia [73]. A 68-year-old Chinese woman with anti-IFNγ auto-antibodies suffered from recurrent herpes zoster infections in addition to an NTM infection [74]. In a group of 83 patients with postherpetic neuralgia, 12% showed high-titer neutralizing anti-IFNγ auto-antibodies [75]. A 45-year-old-male with anti-IFNγ auto-antibodies and recurrent TB also suffered from recurrent herpes zoster episodes [12]. In a cohort of 28 patients with NTM infections, about a quarter also suffered from oral herpes simplex virus, and a quarter from herpes zoster [63]. In a cohort of 45 patients with NTM infections and anti-IFNγ auto-antibodies, 62% also had a history of herpes zoster infection [24]. Among a Taiwanese cohort of 17 patients with disseminated NTM infection and anti-IFNγ auto-antibodies, 71% also suffered from herpes zoster [66]. Interestingly, in the largest cohort study on anti-IFNγ auto-antibodies to date, 5/52 patients with disseminated NTM infection and 10/45 patients with opportunistic infections with or without NTM infection (both groups with the highest titers of these antibodies) also suffered from local VZV infections and 3/45 in the latter group also had disseminated VZV infection [13].

Furthermore, cytomegalovirus has also been described as a coinfection with NTM infections in two Asian women with neutralizing anti-IFNγ auto-antibodies [42,76]. All in all, the exact prevalence of ACAAs in herpes virus infections is unknown, but given the presented case reports and patient cohorts, there seems to be a possible correlation with auto-antibodies directed against IFNγ, a correlation that should be further assessed in larger cohorts.

### 3.2. Human Immunodeficiency Virus (HIV)

Little research has been performed on ACAAs in HIV, whereas it could be an interesting topic for future research with regards to their influence on HIV progression and as an additional risk factor for (opportunistic) infections. Thus far, it has been shown that anti-TNF-α auto-antibody titers are significantly higher in slow/non-progressing people living with HIV (PLHIV) than in seronegative healthy controls and correlate positively with viral load and CD8+ cell count; they correlate inversely with the CD4+ cell count, however. Whether there is an association with certain opportunistic infections is unknown [77]. In an observational study, anti-IFNγ auto-antibodies have been shown in PLHIV. However, further assessment did not show any neutralizing capacity [78]. In contrast, in a follow-up study, in 40 asymptomatic PLHIV with a CD4 count > 400 cells/mm^3^, all of whom were intravenous drug users, showed that the anti-IFNγ auto-antibodies in these patients did have a high neutralizing capacity [79]. However, whether these ACAAs play a role in opportunistic infections or the immunological defense against HIV remains unknown.

### 3.3. Severe Acute Respiratory Syndrome Coronavirus 2 (SARS-CoV-2)

During the recent coronavirus disease 2019 (COVID-19) pandemic, several studies to determine the role of ACAAs in this infection have been published, mainly pointing towards a role for auto-antibodies directed against type I interferons, but also ACAAs to other cytokines such as IL-12, IL-17, and IL-22 were found [80,81,82]. It was shown that auto-antibodies against IFNα and IFNω (and not IFNβ) were present in up to 20% of life-threatening COVID-19 infections and COVID-19 deaths (in comparison to less than 5% of healthy controls), and were more common in elderly male patients [83,84,85,86,87]. The presence and concentration of ACAAs correlated with the disease severity [84,85] and anti-type I IFN auto-antibodies were also present in the nasal mucosa of COVID-19 [88,89]. In a Spanish cohort of 46 patients with severe COVID-19, autoantibodies to IFNα and IFNω were found in 10%. As no anti-IFNβ auto-antibodies were found, these patients were treated with IFNβ supplementation. This did, however, not result in faster clinical improvement [90]. In an American clinical trial, the effect of adding IFNβ to COVID-19 treatment (but not specifically in patients with ACAAs) was assessed, but in this trial, too, without clinical effect [91].

As there appears to be a central role of type I interferons in the immune defense against COVID-19, four patients with a severe infection in the ICU were treated with plasmapheresis. It was shown that concentrations of anti-IFNα auto-antibodies were effectively reduced in blood and tracheal aspirates. Two of these patients died and two patients survived [92]. Also, a single patient with autoimmune polyendocrine syndrome type 1 (APS1) with known neutralizing auto-antibodies to IFNα was treated with plasmapheresis, which resulted in the reduction of autoantibodies [93]. However, as patients with APS1 often have autoantibodies to type I interferons and not all APS1 patients develop severe COVID, other factors than auto-antibodies against type I interferons in determining COVID-19 severity seem likely [94]. Finally, as no controlled trials have been performed to assess the role of plasmapheresis (or other anti-ACAA therapy) in COVID-19, this remains a field for further research.

In the multisystem inflammatory syndrome in children (MIS-C), which occurs in a small group of children after COVID-19, the effect of ACAAs was determined. It was reported that in 21 children with MIS-C, 62% showed auto-antibodies against IL-1 receptor antagonist (IL-1RA) [95], which could explain the severe proinflammatory reaction in these patients. In another study, mainly anti-IFNγ auto-antibodies (in more than 80% of patients) were found, followed by auto-antibodies directed against IL-17, IFNα, IL-6, and IL-22 [96]. However, in a third study, no anti-IFNα auto-antibodies were found in 199 MIS-C patients [97]. Hence, major differences in findings between studies render it difficult at this point in time to determine whether there is a causative role of ACAAs in MIS-C.

### 3.4. Other Respiratory Infections

In 267 patients from five different cohorts with severe respiratory symptoms (due to viral, bacterial, or fungal infection, or non-infectious critical illness) that screened negative for COVID-19, more than 50% demonstrated ACAAs. The majority of ACAAs were found in patients with an infection, of whom the majority were infected with influenza. The most frequently found neutralizing ACAAs were directed against type I interferons, with anti-GM-CSF and anti-IL-6 auto-antibodies in second and third place, respectively [98].

## 4. Fungal Infections

### 4.1. Candida

The role of ACAAs has most extensively been studied in patients with chronic mucocutaneous candidiasis (CMC) where auto-antibodies against IL-17 and IL-22 play a role. CMC can be caused by several underlying disease processes and mutations, but ACAAs are most commonly seen in autoimmune polyendocrinopathy-candidiasis-ectodermal dystrophy (APECED, also known as APS1). APS1 is caused by mutations in the autoimmune regulator (*AIRE*) gene, resulting in the loss of thymic deletion of autoreactive T cells with multiple autoimmune features such as endocrinopathies as a result. Neutralizing autoantibodies against IL-17A (41%), IL-17F (75%), or IL-22 (91%) were found in a cohort of over 150 APECED patients. In addition, neutralizing antibodies against type I interferons were found in APECED patients. Antibodies were mainly found in patients with CMC [99,100,101]. ACAAs were infrequent in patients with other genetic causes of CMC [102]; however, one study also reports IL17F autoantibodies in 11 out of 17 patients with a signal transduction and activator of transcription (*STAT*) 1 gain of function mutation [103]. Especially anti-IL-17A auto-antibodies seem to be correlated with the highest risk of CMC [104]. One study in 17 patients with thymic malignancies found that 12 demonstrated ACAAs directed against one or more cytokines, including type I IFN, IL-12, IL-17A, B cell activating factor (BAFF), IL-1α, TNF-α, IL-6, IL-18, A proliferation-inducing ligand (APRIL), and C-C chemokine receptor 7 (EBI1/CCR7). Five of these patients had opportunistic infections (three with CMC, two with disseminated VZV, one with disseminated cryptococcosis and one with additional infections with *Scedosporium apiospermum* and *M. avium*). Interestingly, the two patients with the highest titers of anti-IL-17A auto-antibodies and whose antibodies had neutralizing activity, had CMC [105]. Proven treatment is currently limited to intermittent azole therapy. Experimental data have been gathered about intravenous immunoglobulin (IVIG) therapy, but no trials have been performed in patients with ACAAs and CMC [106]. Experience with other immunomodulatory therapies such as rituximab or cyclophosphamide has not been reported.

### 4.2. Cryptococcus

Cryptococcal infections, comparable to *Nocardia* infections, have mainly been associated with anti-GM-CSF auto-antibodies [16,107,108]. However, one paper reported 10 non-HIV positive patients with cryptococcosis and disseminated NTM infection, in whom anti-IFNγ auto-antibodies were shown [109]. Also, in patients with anti-IFNγ auto-antibodies with or without coinfections with NTM, cryptococcal infections are sometimes shown [13,24,25,109,110,111,112]. In a Taiwanese cohort of 39 otherwise healthy patients with pulmonary, cerebral, or disseminated cryptococcosis, high titers of anti-GM-CSF auto-antibodies were found in 15 patients, 14 of whom presented with central nervous system (CNS) involvement. Interestingly, in 11 out of these 14, the causative microorganism was found to be *Cryptococcus gattii* [113]. In another Taiwanese study, slightly lower numbers of anti-GM-CSF auto-antibodies were found: out of 23 HIV-negative patients, two with disseminated (including CNS infection), three with exclusively pulmonary, and one patient with exclusively musculoskeletal involvement were found to have anti-GM-CSF autoantibodies. Five out of five patients with auto-antibodies and speciation of *Cryptococcus* available were shown to be infected with *C. gattii*. ACAA serum concentrations were higher in the three patients with extrapulmonary cryptococcosis than in those with exclusively pulmonary infection [114]. Several case reports or small case series with cryptococcal disease in over 30 patients with GM-CSF autoantibodies are currently available in the literature. *C. gattii* infections are more frequently reported than infections due to *C. neoformans*. Interestingly, reported cases frequently demonstrate cerebral involvement [17,115]. No reports on additional immunomodulatory treatment are available from the literature. Although limited long-term follow-up data are available for patients with anti-GM-CSF auto-antibodies, it has been suggested that they should be counseled for the development of PAP later in life (although the exact risk is unknown) [116]. Whether these patients should receive prophylactic treatment remains unknown, although recurrence was relatively rare in the few papers with longer follow-up data [114,115].

### 4.3. Histoplasma

Infection with the dimorphic endemic fungus *Histoplasma capsulatum* has mainly been associated with anti-IFNγ auto-antibodies, although three cases in patients with PAP (and, therefore, likely positive for anti-GM-CSF auto-antibodies) have been reported as well [117]. In addition, in a murine study with pulmonary *Histoplasma* infection, mortality was approximately four times higher in mice treated with anti-GM-CSF antibodies [118]. Most of the patients with anti-IFNγ autoantibodies and histoplasmosis also presented with an NTM infection [13,25,111,112], but some reports also concern solitary *Histoplasma* infection [9,119,120]. No systematic analysis of cases is available and no data on recurrence and the potential beneficial effect of prophylactic treatment are available.

### 4.4. Talaromyces

*Talaromyces* (formerly *Penicillium*) *marneffei* is a dimorphic fungus, which is mainly endemic in Southeast Asia. This fungal infection is seen in advanced HIV infections but has also been described in patients with anti-IFNγ auto-antibodies with or without NTM infection [13]. In as case series of eight Chinese patients with anti-IFNγ auto-antibodies, all eight showed serological evidence of (previous) *Talaromyces* infection [62]. In a cohort of 58 HIV-negative patients with severe talaromycosis, in 55 anti-IFNγ auto-antibodies were found. No neutralizing auto-antibodies against GM-CSF, IL-6, IL-17A, IL-12, and IL-23 were found [121]. In another cohort of 42 patients with talaromycosis, 22 were positive for anti-IFNγ auto-antibodies [122]. In patients with positive anti-IFNγ auto-antibodies, disseminated disease was more common, as well as coinfections with other pathogens (such as NTM) and they demonstrated a higher mortality [24,39,122,123,124,125]. In two systemic reviews of patients with anti-IFNγ auto-antibodies, 18.3%, respectively, 16.3% of these patients with anti-IFNγ auto-antibodies were diagnosed with talaromycosis [20,23], the second most common infection after NTM infections. In one study, five patients with anti-IFNγ auto-antibodies and therapy-refractory talaromycosis were treated with cyclophosphamide. All patients were able to stop antimicrobial therapy 3–12 months later, with only one relapse during a 2-year follow-up time [126]. No other studies on the potential role of immunomodulatory therapy are available at this time.

## 5. Parasitic Infections

Data on ACAAs in parasitic infections are limited. There are two case reports that describe cerebral toxoplasmosis in patients with anti-IFNγ autoantibodies. One reports on a German patient who also suffered from infection with *Mycobacterium avium* complex and *Salmonella* sepsis [61], and the other on a Japanese patient with an additional disseminated NTM infection and recurrent herpes infections [127]. In a murine toxoplasmosis infection model, it was shown that treatment with anti-IFNγ antibodies resulted in increased brain and lung parasitic loads compared to control [128]. In a murine model for African trypanosomiasis, it was shown that during the infection anti-IFNγ autoantibodies were induced. Whether this also occurs in humans, whether these auto-antibodies are functionally neutralizing, and whether the natural existence of anti-IFNγ auto-antibodies would result in other disease outcomes remains elusive [129].

## 6. Discussion

In this review, we provide an overview of the role of different ACAAs in several infectious diseases. However, although many case reports and patient cohorts have been presented, many important questions remain, both diagnostic and therapeutic (Table 1). In some cases, the causative role of ACAAs in disease appears more straightforward (e.g., anti-IFNγ auto-antibodies in NTM infections), but in most infections, it remains unknown whether the ACAAs are indeed the cause or a result of the infection and how big their role exactly is. Also, long-term data on the natural course of these ACAAs are lacking (Table 1). Furthermore, not for all ACAAs, the neutralizing capacity has been unequivocally demonstrated. Whether the titer of ACAAs is predictive of their effect remains elusive in most cases, just as whether ACAAs only play a role in the dissemination of an established infection or are already at the time of acquiring the primary (most often pulmonary) infection. In several case reports and case series, immunomodulatory therapies, such as rituximab or cyclophosphamide, have been employed, in many cases with good effect, but until now not in a randomized and controlled fashion. However, this could be the result of biased publishing, with only the presentation of cases with a positive outcome. Furthermore, it is yet unknown whether patients with ACAAs would benefit from secondary prophylaxis or extended consolidation treatment after completing initial antimicrobial treatment, which are essential questions to be answered. Last, the main focus in ACAAs has thus far been on ACAAs against proinflammatory cytokines, but one of the studies on MIS-C in COVID-19 also shows a role for antibodies against IL-1RA, an anti-inflammatory cytokine. ACAAs against anti-inflammatory cytokines could therefore also play a role in hyperinflammatory reactions in certain infections and should also be a topic of future research.

In order to address the questions above, two issues are of major importance for current clinical practice. First, testing for ACAAs should become more readily available, making it easier to diagnose these conditions. Currently, it is highly likely that many diagnoses of ACAAs are missed due to limited diagnostic availability and limited knowledge about ACAAs in general. Furthermore, testing for ACAAs in clinical circumstances where opportunistic infections are expected, such as in advanced HIV or patients with immune suppressive therapy, should be performed more frequently as well. Potentially, ACAAs could at least constitute an additional risk factor for these infections under these circumstances. Second, ideally, all patients diagnosed with ACAAs should become part of an international study cohort. By improving diagnostic availability, these cohorts will become larger over time. Long-term observation will shine a light on the natural course of ACAAs, the risk for recurrence of disease, and the need for prophylaxis. Moreover, large patient cohorts could make stratified therapeutic intervention studies possible.

## Figures and Tables

**Figure 1 ijms-25-00515-f001:**
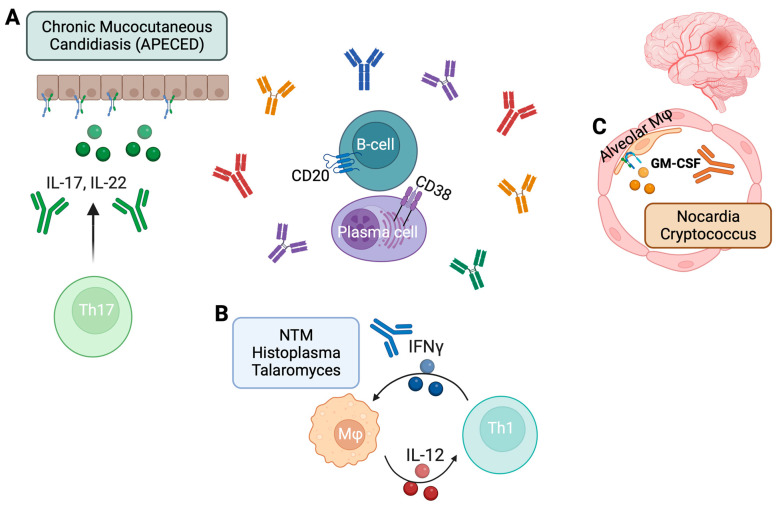
Overview of most important ACAAs and their suggested mode of action. Central to the process of auto-antibody production are B lymphocytes and plasma cells, which can be targeted with host-directed therapies, such as rituximab (anti-CD20). (**A**) The cytokines IL-17 and IL-22 are essential in the interaction between epithelial barrier and *Candida*. IL-17 induces production of antimicrobial peptides and IL-22 enforces epithelial cell proliferation and repair, both through the IL-17-receptor and IL-22-receptor present in epithelial cells. This process is disrupted by anti-IL-17 and anti-IL-22 auto-antibodies. (**B**) IFNγ produced by Th1-cells is an essential activator of macrophages. It improves killing of intracellular pathogens, such as NTMs and several fungi, and induces IL-12 production, which in turn stimulates IFNγ production by Th1-cells. Anti-IFNγ auto-antibodies disrupt this proinflammatory loop. (**C**) GM-CSF is essential in the activation of, among others, alveolar macrophages, and the induction of production of reactive oxygen species. Anti-GM-CSF auto-antibodies therefore increase the risk for specific pulmonary infections with *Nocardia* and cryptococcal species and apart from local infection also increase the risk for disseminated disease such as *Nocardia* brain abscesses and cryptococcal meningitis. APECED, Autoimmune polyendocrinopathy-candidiasis-ectodermal dystrophy; GM-CSF, Granulocyte-macrophage colony-stimulating factor; IFN, Interferon; IL, Interleukin; Mφ, macrophage; NTM, Non-tuberculous mycobacteria; Th, T-helper cell.

**Table 1 ijms-25-00515-t001:** Overview of ACAA targets and current evidence. ACAA, Anti-cytokine auto-antibody; APECED, Autoimmune polyendocrinopathy-candidiasis-ectodermal dystrophy; COVID-19, Coronavirus disease 2019; GM-CSF, Granulocyte-macrophage colony-stimulating factor; HIV, Human immunodeficiency virus; IFN, Interferon; IL, Interleukin; NTM, Non-tuberculous mycobacteria.

Disease	ACAA Target	Evidence	Future Directions for Research
Bacterial			
Tuberculosis	IFNγ	Case reports	Define role in larger cohorts
	GM-CSF/IL-12	Case reports	Define role in larger cohorts
NTM	IFNγ	Proven in larger cohort studies	Therapeutic interventions
	GM-CSF	Case reports	Define role in larger cohorts
*Nocardia*	GM-CSF	Clear association in PAP patients Only case reports in non-PAP patients	Define role in larger cohorts and provide long-term follow-up data
Viral			
Herpes simplex/zoster	Type I and II IFNs	Multiple case reports	Define role in larger cohorts
HIV	IFNγ	Two older cohort studies	Evaluate ACAAs in a new cohort
COVID-19	IFNα and -ω	Large cohort studies No effect of IFNβ supplementation	Therapeutic interventions
Fungal			
*Candida* (in APECED/APS1)	IL-17/IL-22/IFNα	Shown in several cohort studies	Therapeutic interventions
Cryptococcosis (especially *C. gattii*, but also *C. neoformans*)	GM-CSF	Shown in small cohort studies	Define role in larger cohorts and provide long-term follow-up data
	IFNγ	Case reports	Define role in larger cohorts
Histoplasmosis	IFNγ/GM-CSF	Case reports	Define role in larger cohorts
Talaromycosis	IFNγ	Proven in cohort studies	Therapeutic interventions
Parasites			
Toxoplasmosis	IFNγ	Two case reports	Define role in larger cohorts

## Data Availability

Not applicable

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
