# Peer review of "Anticytokine Autoantibodies in Infectious Diseases: A Practical Overview"

_ijms, 2023, doi:10.3390/ijms25010515_

Round 1

Reviewer 1 Report

Comments and Suggestions for Authors

     In this review, the authors summarize the information available about anti-cytokine autoantibodies (ACAAs) in infectious diseases. Information is presented by infectious disease, including their potential pathogenic roles and potential treatment options. This is an interesting and well-written review critically discussing the available information, pointing out the many unanswered questions in the field and suggesting future research. Thus, this review makes and important contribution to the field. There are only a few minor issues to correct:      

1.      Define IVIG the first time it appears (line 131).

2.      The table at the end of the manuscript is very useful but lacks a title.

3.      A few typos:

-          Line 189:  Nocardia (capital N)

-          Line 350: IL-18 (capital L)

-          Line 371: out (small o)

-          Line 402: Talaromyces (capital T, italics)

Author Response

We would like to thank the reviewer for the positive appraisal of our review and the useful comments. We have adjusted all the issues raised by the reviewer.

Reviewer 2 Report

Comments and Suggestions for Authors

The review article is well compiled and touched many infectious diseases. 
Overall the article is well written. 

please see the following suggestions 

The first lines of abstract and introduction are similar try to change in any of the sections to avoid repetition. 
Additionally, there is no figure in the review article and I suggest to incorporate a well defined figure such as explaining the mechanism by which Anticytokine autoantibodies play an important role in infectious diseases. 
There mechanism of action can be considered 

and they have been produced can also be considered. 

Best Wishes 

Comments on the Quality of English Language

The quality of English language is fine and acceptable. 
However, final checks required to avoid and spelling and grammatical errors.

Author Response

We would like to thank the reviewer for his/her suggestions. We have now changed the first lines of the introduction in order to avoid repetition. Furthermore, we have added a figure to the manuscript to, in a very broad way, explain the mode and site of action of the most important ACAAs. 

Reviewer 3 Report

Comments and Suggestions for Authors

The authors tried to review anticytokine auto-Abs (ACAAs) in various infectious diseases. The aim for this review is interesting and timely. However, systematic errors were found in the draft manuscript.

1. In the draft manuscript, no description regarding molecular mechanisms of the ACAAs production in infectious diseases were found. I think that this is essential, even if this manuscript is clinical paper.

2.  The draft was merely a list of cases. I think that discussion is needed to give the readers deeper insights.       

Author Response

We would like to thank the reviewer for her/his appraisal of our manuscript and the suggestions to improve the manuscript. 

Ad 1. We indeed didn't include any molecular mechanisms of ACAAs in our manuscript as they are not very well defined yet and especially because recently a few very well written reviews specifically on that topic have already been published elsewhere. However, we do agree with the reviewer that it would help the reader to better understand this topic when we would touch upon this issue. Also according to the suggestion of reviewer 2, we have now included a figure that displays the most well-known mechanisms and sites of action of ACAAs with an additional explanation in the figure legend. 

Ad 2. We agree with the reviewer that our review is a list of cases, because we feel that such an overview was missing from literature. Most published reviews focus on the different ACAAs and their effects, but from a clinical approach it is of more assistance to take the disease as a starting point. We therefore provided this overview to display on the one hand which ACAAs play a role in which infectious disease and on the other hand (even more importantly) which therapeutic options have been tried and where evidence is lacking. 

We feel that we already addressed most discussion points in our conclusions paragraph. However, to enhance the discussion on the existing points of debate and areas where evidence is lacking, we have restructured the conclusions paragraph to a discussion paragraph. Additionally, we have rephrased and included several new sentences to provide better support for the subsequent critical stages of research outlined in table 1, with the primary objective to display where current research should aim at in a comprehensive and informative manner.

Round 2

Reviewer 2 Report

Comments and Suggestions for Authors

The authors have addressed the comments sufficiently.

Comments on the Quality of English Language

The English language is acceptable.

Reviewer 3 Report

Comments and Suggestions for Authors

The authors well addressed for all comments by the Reviewer. Thus, I recommend to the Editor that the revised manuscript is now suitable publication in this journal.